# Caries Prevalence Associated with Oral Health-Related Behaviors among Romanian Schoolchildren

**DOI:** 10.3390/ijerph18126515

**Published:** 2021-06-17

**Authors:** Ruxandra Sava-Rosianu, Guglielmo Campus, Anamaria Matichescu, Octavia Balean, Mihaela Adina Dumitrache, Patricia Ondine Lucaciu, Luminita Daguci, Magda Calina Barlean, Laurentiu Maricutoiu, Mariana Postolache, Ramona Dumitrescu, Daniela Jumanca, Atena Galuscan

**Affiliations:** 1Translational and Experimental Clinical Research Centre in Oral Health, Department of Preventive, Community Dentistry and Oral Health, University of Medicine and Pharmacy “Victor Babes”, 300040 Timisoara, Romania; sava-rosianu.ruxandra@umft.ro (R.S.-R.); balean.octavia@umft.ro (O.B.); jumanca.daniela@umft.ro (D.J.); galuscan.atena@umft.ro (A.G.); 2Department of Restorative, Preventive and Pediatric Dentistry, University of Bern, Freiburgstrasse 7, 3012 Bern, Switzerland; guglielmo.campus@zmk.unibe.ch; 3Department of Surgery, Microsurgery and Medicine Sciences, School of Dentistry, University of Sassari, Viale San Pietro 3/c, 07100 Sassari, Italy; 4Oral Health and Community Dentistry Department, Faculty of Dental Medicine, UMP “Carol Davila”, 020021 Bucharest, Romania; mihaela.dumitrache@umfcd.ro; 5Department of Oral Health, “Iuliu Hatieganu” University of Medicine and Pharmacy, 400012 Cluj-Napoca, Romania; ondineluc@yahoo.com; 6Department of Prosthodontics, Faculty of Dentistry, University of Medicine and Pharmacy, 200585 Craiova, Romania; luminita.daguci@umfcv.ro; 7Department of Oro-Dental Diagnosis and Geriatric Dentistry, Faculty of Dental Medicine, “Grigore T. Popa” University of Medicine and Pharmacy, 700115 Iași, Romania; magdabarlean@gmail.com; 8Department of Psychology, West University of Timisoara, 300223 Timisoara, Romania; laurentiu.maricutoiu@e-uvt.ro; 9Department of Program Implementation and Coordination, Romanian Ministry of Health, 010024 Bucharest, Romania; mariana.postolache@ms.ro

**Keywords:** dental caries, prevalence, severity, oral health, behavior, children

## Abstract

Oral health-related behaviors and living conditions play an important role in general and oral health. This study aimed to evaluate caries prevalence and severity in schoolchildren residing in rural and urban areas of Romania, and to correlate these with oral health-related behaviors. An estimation of the required sample size was conducted (sampling error of ±3% at a 95% confidence level), followed by the stratification of administrative units and then the selection of 49 schools. The Hurdle approach was used to analyze the dataset, requiring two sets of analyses for each outcome variable: a multilevel binary model to predict prevalence, and a multilevel Poisson analysis using only non-zero values. The mean and standard deviation (SD) for the dentinal caries index was 4.96 (5.33). Girls were more likely to have non-zero restoration codes (β = 0.14, SE = 0.08, *p* < 0.05). Low education levels of each parent were associated with an increased likelihood of having non-zero carious tooth surfaces (β = 0.23, SE = 0.06, *p* = 0.01; β = 0.22, SE = 0.06, *p* < 0.01). The presence of cavities was predicted by the consumption of carbonated soft drinks (β = 0.19, SE = 0.07, *p* < 0.01), candies (β = 0.13, SE = 0.06, *p* < 0.01), sweetened milk (β = 0.12, SE = 0.06, *p* < 0.05), tea (β = 0.16, SE = 0.08, *p* < 0.05), or cocoa (β = 0.13, SE = 0.06, *p* < 0.05). Furthermore, the non-zero values of the dentinal caries index were more likely in rural schools (β = −0.37, SE = 0.11, *p* < 0.01), and a negative association between the county development index and the fillings/restorations index (β = −0.01, SE = 0.01, *p* < 0.05) was also established. The outcome of this research highlights that the presence of caries (dentinal caries index) in Romanian schoolchildren is influenced by their socioeconomic background, as well as their specific consumption behaviors.

## 1. Introduction

Caries is a multi-factorial disease resulting from a series of events that evolve in a chain, lasting for years [1,2]. It is still the single most common non-communicable disease worldwide, and it is not self-limiting or treatable with antibiotics. Between 1999 and 2004, the prevalence of untreated caries lesions was 24.5% for children aged 6–11, and 19.6% for adolescents aged 12–16 years [3], leading to difficulties in eating and sleeping, pain, the need for restorative treatment, emergency visits and inpatient hospitalizations, as well as a poor quality of life. Public health efforts, aimed at addressing the pediatric caries epidemic, have mostly focused on tooth level interventions [4], and have led to a decreasing trend in most industrialized countries [5]. The widespread use of fluoridated toothpastes and preventive programs are the main factors involved in this decrease [6,7]. 

In order to benefit from a more comprehensive understanding of how social factors might influence the prevalence of caries, the differences observed between rural and urban areas also constitute a strong factor to be considered. Rural living deprivation has been linked with poorer dental health in several surveys [8,9,10]. 

Numerous comprehensive studies in the field have been conducted in countries like Italy, Greece, Hungry, Slovenia, and Croatia, and these have provided valuable insights and future considerations, reporting big differences among countries (i.e., DMFT equal to 4 in Croatia; 4.5 in Slovenia; 3.8 in Hungary; 2.05 in Greece; 0.8 in Italy) [8,10,11,12,13].

Despite multiple studies being conducted in neighbouring countries, Romania lacks comprehensive national oral health surveys in children or adults. A previous study by Petersen et al. was conducted in the 1990s covering only five major cities and without taking into account rural areas. Some local papers describe a high caries prevalence [14,15]. Only one longitudinal study assesses caries trends in Romanian schoolchildren [16]. A contemporary assessment is therefore mandatory in order to advance scientific understanding and allow for the development of adequate public health policies.

A cross-sectional epidemiological survey “Romanian Oral Health Survey” was designed and conducted in 2019–2020, focusing on two age groups, and this paper was aimed to describe the caries prevalence and severity of carious lesions in 11–14-year-old schoolchildren residing in rural and urban areas of Romania, and to correlate these findings with oral health-related behaviors.

## 2. Materials and Methods

### 2.1. Study Design and Sample Selection

The first phase of the research represented the sample selection of the children and the development and validation of questionnaires for the targeted samples. 

The public schools to be included in the survey were selected based on the number of children registered by the National School Inspectorate in the 42 counties of Romania. Romania is divided into administrative units called counties. Each county is administrated by a major city and gathers together several smaller cities and villages. Rural and urban localities are categorized by the National Administration according to the number of inhabitants. As Romania doesn’t benefit from any source of systemic fluoridation, data regarding fluoridation have never been gathered. Moreover, there are no water or milk fluoridation programs, and sources of fluoride in the soil and water are scarce [17].

An a-priori estimation of the required sample size was conducted assuming a sampling error of plus or minus 3% at a 95% confidence level. 

The number of children enrolled in the 8th grade National Exam for each Romanian school was established using public information available on the websites of all County School Inspectorates (41 counties plus the Romanian Capital). This process involved the evaluation of publicly available information from 4696 schools, which aided our understanding of the territorial distribution of the target population. The sample was stratified and randomized. The stratification was performed on the administrative units (counties), and on locality type (i.e., urban versus rural localities). For each county, the total number of pupils was determined and expressed as a percentage share related to the total number of children. The percentage share was used to estimate how many children would have to be included in each county. The obtained number was then divided based on locality type (i.e., urban versus rural) and the final target number of evaluations was obtained. MS Excel’s randomization function was used to select an urban and a rural school for each county. As a result, a total of 49 schools distributed in rural and urban areas were selected. At every school, all the students in one class were invited to participate in the study. Although the participation rate was not recorded in detail, it is possible to estimate an inclusion of at least 90% of all children in these 49 schools. 

School level predictors included also the County Developmental Index, which is a sociological index that combines county-level variables: education stock, life expectancy at birth, medium age of adult population, average living space, number of private cars to 1000 inhabitants, and average household gas consumption. The aggregation of all these different variables was done using factor analysis, and the scores were relevant for assessing the county’s workforce potential, and also the county’s economic potential. 

Data was collected using the Oral Health Questionnaire for Children developed by the World Health Organization and described in the WHO Oral Health Surveys—Basic Methods, 5th edition, 2013 [18]. The English version of this questionnaire was translated by two independent translators, and the differences were settled in a face-to-face meeting. Furthermore, two experts were consulted (i.e., Ph.D. in educational and in developmental psychology, respectively) regarding the readability of the Romanian version of the Oral Health Questionnaire for Children. It comprises questions regarding general information, age (date of birth and age in years), educational level of the parents/caregivers, experience of pain related to teeth, dental visits, frequency and aids for tooth cleaning, and consumption of sugary foods and drinks. Questionnaires were submitted to the children one week prior to the clinical examination and filled in by the children in collaboration with their parents/caretakers at home. Positive informed consent had to be filled in by the parents/caretakers. The questionnaires were gathered by the examiners on the day of examination. 

### 2.2. Clinical Examination

Children were examined in a classroom or any available room, with the teacher present during the procedure, using special lighting sources and examination kits. Each participant came to the examination with the informed consent and questionnaire signed by parents/caregivers. ICDAS criteria were used to classify visual caries lesion severity, and the ICCMS Guide for Practitioners and Educators [19] was used to classify the presence of filling material in all of the available tooth surfaces of permanent and primary teeth. Cotton rolls were used to remove dental plaque or food debris. ICDAS codes for every surface were recorded on a specific chart. ICDAS codes 1 and 2 were recorded as “A” because air drying was not possible. The examination chart was attached to the questionnaire and informed consent of each participant.

Examination was conducted by 10 calibrated dentists. The calibration procedure was carried out by examining 21 subjects displaying an inter-examiner kappa ranging from 0.74–0.86 and intra-examiner kappa ranging from 0.81–0.92. The kappa coefficient for fillings was consistently in the excellent range, while for lesion severity it was in the good to excellent range. None of the children included in the present sample were re-examined. The required calibration of the dentists was performed prior to the data collection, as described.

### 2.3. Data Analysis

ICDAS codes higher than 3 (caries into dentin) were computed as dentinal caries (number of carious tooth surfaces), missing teeth due to complex carious lesions as MT, and number of restored tooth surfaces as FT. By computing these indices, the variables associated with caries (the dentinal caries index) and the variables associated with the treatment of decayed teeth (the MT and FT indices) were analyzed. As children were sampled from 49 schools, a multilevel approach was used to analyze the data. The Hurdle approach was used to analyze the dataset for each outcome variable. The Hurdle approach requires two sets of analyses for each outcome variable: a multilevel binary model (multilevel logistic regression using the Bernouli estimation) to predict the prevalence of each outcome, and so to predict the absence or the presence of caries using the entire sample, and a multilevel Poisson analysis using only the children with non-zero values. Because more than 90% of the sample did not have any missing teeth due to extractions, the Poisson analysis on this outcome was not conducted, as the remaining sample would have been too small (*n* = 57). Regarding the remaining outcome variables, the over-dispersed multilevel Poisson analysis was used for the dentinal caries index (mean dentinal caries = 4.96, SD dentinal caries = 5.33) and the classical multilevel Poisson analysis for the FT index (mean = 3.95, SD = 3.37). Because the interpretation of multilevel analyses is eased if the predictors are centered, the median value was used to center each predictor. The analyses focused on studying the relationships between each predictor and the three output variables. For each predictor, two separate regression analyses were conducted: a multilevel logistic regression and a multilevel Poisson regression. These analyses accounted for the nested nature of the data and are equivalent to examining the zero-order correlations between predictors and outcomes. For all multilevel analyses, the HLM version 7.02 was used.

The potential confounders have not been adjusted because, as this is the first nationally representative study on Romanian children, the research was interested in the relationships between demographics and the oral health parameters. Moreover, by adjusting for potential confounders, partial associations between oral health-related behaviors and oral health parameters would have had to be reported. As the partial effect is dependent on the number, the operationalization, and the quality of the controlled variables, it proves difficult to compare it to other studies. Lastly, most of the oral health-related behaviors were not significantly associated with demographic or socioeconomic variables; therefore, controlling for the later variables has little impact on the relationship between oral health behaviors and oral health parameters. 

## 3. Results

Gender distribution was similar within the sample (47.37% boys and 52.01% girls). A total of 43.71% of the children resided in rural areas and 56.29% in urban areas, mostly small cities. Almost one third of the parents had high school education (Table 1). 

Figure 1 shows the distribution of caries-free surfaces (ICDAS codes 0) among the studied population. 

ICDAS caries codes and restoration codes differed by region (Table 2) and gender (Table 3). The percentage of caries-free children in Oltenia (south) was 17.81% in rural regions and 26.32 in urban regions, whereas in the north (North-western Transilvania) the percentages were lower: 4.00% in rural regions and 20.00% in urban regions. Additionally, the mean number of dentinal caries was higher in North-western Transilvania: 4.24 (SD 5.44) in rural regions and 2.42 (SD 3.97) in urban areas. The highest number of dentinal caries was seen in the central part of the country (Central Transilvania), with a mean of 6.05 (SD 9.11) in rural areas and 3.92 (SD 6.55) in urban areas. The average number of fillings was higher in the northern part of the country and lower in the south. Moldova, the North-east of the country, had the highest mean number of restorations (4.09; SD 4.17; 4.63, SD 4.64 in rural and 3.48, SD 3.49 in urban areas), and Oltenia (the southern part) had the lowest number of restorations (0.52, SD 1.21; 0.80, SD 1.01 in rural and 1.05, SD 1.72 in urban areas). The capital city, Bucharest, had the highest percent of caries-free surfaces (26.26%), and the western part of the country (Banat) had the lowest percentage of caries free surfaces in the urban area (10.22%). Regarding the severity of the lesions, incipient caries lesions were more frequently seen in the western (Banat) and north-western part of the country (North-western Transilvania), whereas extensive caries lesions were seen in the other regions. 

Girls were more likely to have non-zero restoration codes, in comparison to boys (β = 0.14, SE = 0.08, *p* < 0.05). Low education levels of each parents were also associated with the increased likelihood of having non-zero carious teeth surfaces (β = −0.23, SE = 0.06, *p* < 0.01, for the father’s education level; β = −0.22, SE = 0.07, *p* < 0.01 for the mother’s education level) and non-zero restoration codes (β = −0.14, SE = 0.07, *p* < 0.05 for the father’s education level; β = −0.18, SE = 0.06, *p* < 0.01 for the mother’s education level) (Table 4). Interestingly, Poisson regression analyses indicate that parents’ low education level was associated only with the high number of carious tooth surfaces, not with the number of restored tooth surfaces (Table 5).

In the context of the relationship between the consumption of sweetened products and the outcome variables, a significant association, mostly with the presence of cavities (i.e., non-zero values on the dentinal caries index) can be observed. The presence of cavities was predicted by the consumption of carbonated soft drinks (β = 0.19, SE = 0.07, *p* < 0.01), candies (β = 0.13, SE = 0.06, *p* < 0.01), sweetened milk (β = 0.12, SE = 0.06, *p* < 0.05), sweetened tea (β= 0.16, SE = 0.07, *p* < 0.05), or sweetened cocoa (β = 0.13, SE = 0.06, *p* < 0.05). Interestingly, negative relationships can be observed between the existence of restored tooth surfaces (the FT index), the frequent consumption of carbonated soft drinks (β = −0.15, SE = 0.06, *p* < 0.05), and the frequent consumption of sweetened cocoa (β = −0.13, SE = 0.07, *p* < 0.05). 

The Poisson regression analyses of non-zero count data (Table 5) suggest that the consumption of sweetened products had limited links with the number of cavities or the number of restorations. Significant positive relationships between the frequent consumption of carbonated soft drinks and the dentinal caries index (β = 0.12, se = 0.05, *p* < 0.05), and between the frequent consumption of sweetened cocoa and the FT index (β = 0.09, SE = 0.04, *p* < 0.05) can be observed. A detailed description of nutritional behavior can be found in the Appendix A.

Lastly, it was evaluated whether two school-level variables (place of residence type and the county development index) were also associated with the oral health indices. The results show that the non-zero values of the dentinal caries index were more prevalent in rural schools, in comparison to the schools from urban areas (β = −0.37, SE = 0.11, *p* < 0.01). The other significant relationship can be observed as a negative association between the county development index and the RT index (β = −0.01, SE = 0.003, *p* < 0.05).

## 4. Discussion

According to the objective of our survey, the prevalence and severity of caries was evaluated for the specific age group. Based on the results highlighted above, four categories of predictors explain the variance of the dentinal caries index and fillings/restoration index: personal characteristics of the respondents, hygienic behavior, consumption behavior, and living area characteristics. 

Rural residence appears to have a small effect over and above that of deprivation, both in terms of oral health (prevalence of any caries experience) and in terms of treatment (prevalence of extractions). This could be explained by contextual factors related to the lifestyle and health behavior of the people who live in rural areas [20]. 

Health disparities have profound social, political, and economic consequences for society. Children living in poverty, or poor housing, who have a poor diet or lack access to high-quality early years education, are more likely to experience chronic adult illnesses and the intergenerational perpetuation of poverty and poor health [21,22].

The results regarding the living area characteristics highlight that children living in rural areas have a higher probability of developing caries compared to children from urban areas. However, the restoration index is not affected by the rural–urban residency. Furthermore, the number of restorations is negatively impacted by the level of county development and has no influence on the presence of caries or restoration behaviors. 

Gender, the first variable of the demographic dimension, does not produce any relationship with the presence or absence of caries, as boys and girls appear to have a similar prevalence of caries. Nonetheless, gender is a predictor of restoration behavior, as restoration is more likely to occur in girls. This difference can be explained by a higher level of attention paid by parents, by more requests of girls compared to boys, or by gender differences in attendance to medical services or oral hygiene, leading to more advanced stages of untreated decay in boys’ teeth. The fear factor among girls who prefer to have decayed teeth filled, or even a perceived fear factor from the perspective of the practitioner was also described in Scotland in relation to the presence of restorations [23]. 

The presence of restoration might be influenced by the lack of dental check-ups among low-income families and by previous negative experiences of service provision among both children and caregivers, as well as logistical difficulties in making and keeping appointments as a result of a lack of transportation, an inability to get time off work, or having to balance responsibilities where, for example, single parents have to care for many children [24,25].

The demographic dimension of the respondents included the parents’ level of education (both mother and father). These indicators are important predictors for the absence of caries as well as for restoration behaviors. The higher the level of education of parents is, the greater the probability that the child will not have caries. Similar figures were observed about restoration behaviors. Children with highly educated parents have a higher probability to have restoration treatment. Based on this, it could result that the presence of caries in children might be predicted by their parents’ level of education. Furthermore, parents’ level of education might be a useful predictor not only for the presence or absence of caries, but also for the number of caries that children have. The lower the level of education of parents is, the higher number of caries their children have.

Socio-demographic patterns in caries were highly influenced by a high education level of the father, suggesting that this parameter should affect caries severity, as previously reported [13,25]. An appealing speculation could be that parents with a high educational level could establish better oral health habits in their children.

The survey data show that there is no apparent relationship between the presence of caries and the restoration index of the frequency of tooth brushing. However, there is a statistically significant relationship between the lack of hygienic behaviors and the number of caries lesions. 

Good oral health behaviors reduce the risk of caries and are thought to have been key factors in the reduction of caries in developed countries observed in the past 20 years [25]. Despite this research involving the analyses of multiple types of food, not all of them were found to have a direct relation to the prevalence of caries and restoration indices. Carbonated soft drinks and cocoa intake have a direct proportional link with the presence of dentinal caries index and an indirect proportional relation with the presence of restoration index. This relation highlights the presence of a category of the population for whom neither consumption behaviors nor medical treatment is important. Furthermore, a high frequency of carbonated soft drink consumption is associated with a high number of caries, while a high frequency of cocoa intake is associated with a high number of restorations. Complementary consumption of candies, milk, and tea has a positive correlation with the presence of caries index in our sample. Moreover, the data shows no correlation between the caries and restoration index, and fruit and pastry consumption.

One of the limitations of the study resides in the interpretation of the results, which might be biased because of the study’s design, as, in general, a cross-sectional study measures cause and effect at the same point in time, therefore introducing the problem of temporal ambiguity and an inability to establish causal relationships. Among the limitations of this survey, it is possible to recognize some methodological difficulties in organizing and managing the data collection across the 42 counties, and not registering the participation rate can be considered as one of the limitations of this study. Additionally, it is necessary to take into account that the questionnaire data source may not be reliable enough, specifically in relation to socioeconomic data.

The validity of the findings is supported by the complex process of the sample selection and data analysis as well as by the thorough calibration process. The present study is the first one carried out in Romania that takes into account caries prevalence and also evaluates the role of the main caries risk factors and oral health-related behaviors, using the STEPS approach as suggested by the WHO guidelines. This ensures the possibility of evaluating national trends as well as comparing these to other countries. The STEPS approach advises the collection of data on a regular and continuing basis. 

## 5. Conclusions

Caries prevalence (dentinal caries index) in Romanian schoolchildren is strongly influenced by their socioeconomic background, as well as by their specific consumption behaviours. The level of education of the parents, consumption of carbonated soft drinks, milk, candies, tea, and cocoa, as well as rural–urban area residency, influence the prevalence of caries in Romanian schoolchildren. The number of restorations is impacted by gender, parents’ level of education, and the consumption of carbonated soft drinks and cocoa.

## Figures and Tables

**Figure 1 ijerph-18-06515-f001:**
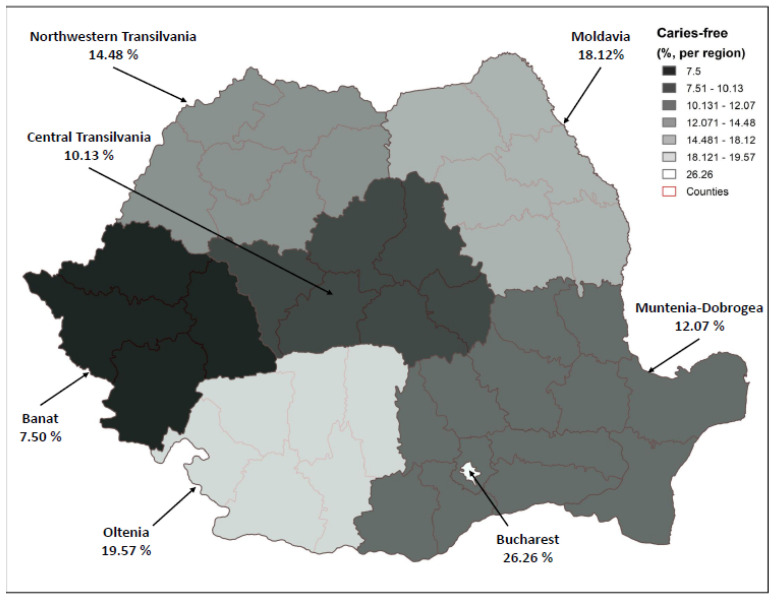
Distribution of caries-free surfaces (ICDAS codes 0) among the studied population.

**Table 1 ijerph-18-06515-t001:** Descriptive statistics of the sample.

Variables	N (%)	Variables	N (%)
**Gender**		**Father’s education**	
Male	388 (47.66)	No school	4 (0.49)
Female	426 (52.33)	Primary school (4th grade)	24 (2.93)
**Age**		Gymnasium (8th grade)	91 (11.11)
11 y.o.	41 (5.01)	Vocational School (10th grade)	124 (15.14)
12 y.o.	514 (62.76)	High School (12th grade)	268 (32.72)
13 y.o.	231 (28.21)	Post-secondary school	29 (3.54)
14 y. o.	24 (2.93)	University studies	149 (18.19)
**Residence**		Don’t know/Don’t answer	106 (12.94)
Rural area	358 (43.71)	I don’t live with a father figure in the house	16 (1.95)
Urban area	461 (56.29)	**Mother’s education**	
**City type**		No school	6 (0.73)
Small town	544 (66.42)	Primary school (4th grade)	20 (2.44)
Middle city	61 (7.45)	Gymnasium (8th grade)	122 (14.90)
Big city	214 (26.13)	Vocational School (10th grade)	113 (13.80)
		High School (12th grade)	234 (28.57)
		Post-secondary school	44 (5.37)
		University studies	181 (22.10)
		Don’t know/Don’t answer	87 (10.62)
		I don’t live with a mother figure in the house	3 (0.37)

**Table 2 ijerph-18-06515-t002:** ICDAS scores (caries and fillings) in the different counties of Romania by living conditions.

	Caries-Free (ICDAS = 0) *(%)*	Enamel Caries (ICDAS 1–3) *Mean (SD)*	Dentinal Caries (ICDAS 4–6) *Mean (SD)*	Fillings Restorations *Mean (SD)*	No Fillings *(%)*
Moldova	18.12%	2.95 (3.10)	2.07 (3.66)	4.09 (4.17)	22.15%
rural	13.75%	3.48 (3.38)	2.41 (4.15)	4.63 (4.64)	17.50%
urban	23.19%	2.33 (2.63)	1.67 (2.97)	3.48 (3.49)	27.54%
Muntenia-Dobrogea	12.07%	4.74 (4.56)	2.18 (3.82)	0.65 (1.44)	75.29%
rural	9.88%	5.57 (4.80)	2.70 (3.90)	0.80 (1.75)	75.31%
urban	13.98%	4.01 (4.23)	1.72 (3.72)	0.52 (1.09)	75.27%
Oltenia	19.57%	2.01 (1.98)	2.50 (4.30)	0.52 (1.21)	78.26%
rural	17.81%	2.15 (1.91)	2.52 (4.38)	0.38 (1.01)	82.19%
urban	26.32%	1.47 (2.17)	2.42 (4.07)	1.05 (1.72)	63.16%
Banat	7.50%	5.60 (5.13)	2.78 (3.88)	1.70 (2.90)	57.50%
rural	3.23%	5.71 (3.58)	3.55 (4.37)	1.39 (2.96)	58.06%
urban	10.20%	5.53 (5.94)	2.29 (3.50)	1.90 (2.87)	57.14%
Central Transilvania	10.13%	4.95 (4.16)	5.08 (8.07)	0.62 (1.45)	78.48%
rural	2.33%	5.51 (4.23)	6.05 (9.11)	0.40 (1.37)	86.05%
urban	19.44%	4.28 (4.03)	3.92 (6.55)	0.89 (1.53)	69.44%
Northwestern Transilvania	14.48%	4.74 (4.54)	3.05 (4.59)	1.84 (3.12)	55.17%
rural	4.00%	6.74 (3.92)	4.24 (5.44)	1.26 (2.12)	68.00%
urban	20.00%	3.68 (4.51)	2.42 (3.97)	2.15(3.51)	48.42%
București-Ilfov	26.26%	3.25 (3.24)	0.53 (1.28)	1.21 (1.89)	54.55%
urban	26.26%	3.25 (3.24)	0.53 (1.28)	1.21 (1.89)	54.55%

**Table 3 ijerph-18-06515-t003:** ICDAS by gender and geographical region.

	Caries-Free (ICDAS = 0) *(%)*	Enamel Caries (ICDAS 1–3) *Mean (SD)*	Dentinal Caries (ICDAS 4–6) *Mean (SD)*	Fillings Restorations *Mean (SD)*	No Fillings *(%)*
Moldova	18.12%	2.95 (3.1)	2.07 (3.66)	4.09 (4.17)	22.15%
male	20.59%	2.71 (3.02)	1.9 (3.66)	3.97 (4.69)	27.94%
female	14.10%	3.26 (3.18)	2.29 (3.71)	4.29 (3.72)	15.38%
Muntenia-Dobrogea	12.07%	4.74 (4.56)	2.18 (3.82)	0.65 (1.44)	75.29%
male	10.34%	4.82 (4.82)	1.97 (3.21)	0.66 (1.49)	75.86%
female	13.79%	4.66 (4.32)	2.39 (4.36)	0.64 (1.39)	74.71%
Oltenia	19.57%	2.01 (1.98)	2.5 (4.3)	0.52 (1.21)	78.26%
male	14.04%	2.07 (2.06)	3.11 (4.92)	0.46 (1.23)	84.21%
female	28.57%	1.91 (1.85)	1.51 (2.82)	0.63 (1.19)	68.57%
Banat	7.50%	5.6 (5.13)	2.78 (3.88)	1.7 (2.9)	57.50%
male	7.69%	5.44 (3.99)	2.64 (3.94)	1.74 (2.96)	53.85%
female	7.32%	5.76 (6.07)	2.9 (3.88)	1.66 (2.88)	60.98%
Central Transilvania	10.13%	4.95 (4.16)	5.08 (8.07)	0.62 (1.45)	78.48%
male	5.56%	5.5 (4.1)	6.53 (10.48)	0.22 (0.72)	88.89%
female	13.95%	4.49 (4.21)	3.86 (5.09)	0.95 (1.8)	69.77%
Northwestern Transilvania	14.48%	4.74 (4.54)	3.05 (4.59)	1.84 (3.12)	55.17%
male	16.67%	4.53 (4.66)	3.71 (4.5)	2.18 (3.7)	53.03%
female	11.69%	4.97 (4.49)	2.48 (4.66)	1.6 (2.54)	55.84%
București-Ilfov	26.26%	3.25 (3.24)	0.53 (1.28)	1.21 (1.89)	54.55%
male	31.43%	2.94 (3.21)	0.23 (0.69)	1.29 (2.22)	68.57%
female	23.44%	3.42 (3.27)	0.69 (1.49)	1.17 (1.7)	46.88%

**Table 4 ijerph-18-06515-t004:** Multilevel regression analysis of individual-level variables and caries figures.

Multilevel Logistic Regressions of Occurrence vs. 0. Non-Occurrence *(Bernouli Estimations)*						
Extracted Teeth		95% C.I.	Dentinal Caries	95% C.I.	Filled Teeth	95% C.I.
Individual-level variables						
Gender (−1 boy, 1—girl)	0.03 (0.12)	[−0.19; 0.25]	0.03 (00.06)	[−0.07; 0.13]	0.14 (0.07) *	[0.01; 0.26]
Father’s education level (centered around high school)	0.06 (0.11)	[−0.15; 0.28]	−0.22 ** (0.06)	[−0.34; −0.10]	−0.14 * (0.07)	[−0.28; −0.0]
Mother’s education level (centered around high school)	0.00 (0.07)	[−0.13; 0.15]	−0.22 ** (0.06)	[−0.33; −0.10]	−0.18 ** (0.06)	[−0.29; −0.05]
How often teeth are cleaned (centered around “once a day”)	−0.01 (0.15)	[−0.3; 0.28]	−0.07 (0.06)	[−0.19; 0.04]	0.08 (0.06)	[−0.04; 0.21]
Fruit consumption (centered around “daily”)	0.15 (0.08)	[−0.02; 0.32]	0.01 (0.07)	[−0.12; 0.15]	0.11 (0.06)	[−0.01; 0.23]
Pastry (centered around “weekly”)	0.08 (0.12)	[−0.16; 0.33]	0.03 (0.05)	[−0.07; 0.13]	−0.00 (0.06)	[−0.13; 0.11]
Carbonated soft drinks (centered around “weekly”)	0.11 (0.12)	[−0.12; 0.35]	0.19 ** (0.06)	[0.06; 0.32]	−0.15* (0.06)	[−0.27; −0.03]
Honey (centered around “monthly”)	−0.01 (0.07)	[−0.16; 0.13]	−0.02 (0.06)	[−0.14; 0.09]	−0.08 (0.05)	[−0.19; 0.0]
Chewing gum (centered around “monthly”)	0.08 (0.11)	[−0.13; 0.3]	0.12 (0.06)	[0.00; 0.24]	−0.04 (0.04)	[−0.14; 0.04]
Candies (centered around “weekly”)	0.12 (0.09)	[−0.05; 0.30]	0.12 * (0.05)	[0.01; 0.23]	0.01 (0.05)	[−0.1; 0.11]
Milk (centered around “monthly”)	−0.05 (0.10)	[−0.25; 0.15]	0.12 * (0.06)	[0.00; 0.23]	−0.00 (0.04)	[−0.11; 0.1]
Tea (centered around “monthly”)	−0.02 (0.07)	[−0.16; 0.11]	0.16 * (0.06)	[0.03; 0.29]	0.06 (0.06)	[−0.06; 0.2]
Cocoa (centered around “monthly”)	0.09 (0.10)	[−0.11; 0.31]	0.12 * (0.06)	[0.00; 0.25]	−0.12 * (0.05)	[−0.24; −0.01]
School-level predictors						
Locality type (−1 Rural, 1 Urban)	−0.16 (0.17)	[−0.5; 0.17]	−0.36 ** (0.11)	[−0.58; −0.1]	0.19 (0.14)	[−0.08; 0.48]
County development index (centered around the mean)	−0.00 (0.01)	[−0.02; 0.02]	−0.01 (0.01)	[−0.03; 0.0]	0.00 (0.01)	[−0.02; 0.02]

* *p* < 0.05; ** *p* < 0.01.

**Table 5 ijerph-18-06515-t005:** Poisson regression analysis of individual-level variables and caries figures.

Poisson Regressions of Non-Zero Count Data				
	Dentinal Caries	95% C.I.	Filled Teeth	95% C.I.
**Individual-level variables**				
Gender (−1 boy, 1—girl)	−0.06 (0.05)	[−0.14; 0.02]	−0.07 (0.04)	[−0.13; −0.0]
Father’s education level (centered around high school)	−0.14 * (0.06)	[−0.26; −0.02]	0.01 (0.04)	[−0.05; 0.08]
Mother’s education level (centered around high school)	−0.11 * (0.05)	[−0.20; −0.02]	−0.01 (0.03)	[−0.07; 0.05]
How often teeth are cleaned (centered around “once a day”)	−0.12 ** (0.04)	[−0.20; −0.03]	−0.01 (0.08)	[−0.17; 0.15]
Fruit consumption (centered around “daily”)	0.03 (0.04)	[−0.05; 0.12]	−0.04 (0.05)	[−0.15; 0.06]
Pastry (centered around “weekly”)	0.02 (0.05)	[−0.08; 0.13]	0.02 (0.05)	[−0.07; 0.13]
Carbonated soft drinks (centered around “weekly”)	0.11 * (0.05)	[0.02; 0.214]	0.01 (0.05)	[−0.09; 0.11]
Honey (centered around “monthly”)	−0.01 (0.04)	[−0.09; 0.06]	0.03 (0.03)	[−0.04; 0.11]
Chewing gum (centered around “monthly”)	0.08 (0.04)	[−0.00; 0.16]	0.00 (0.04)	[−0.08; 0.09]
Candies (centered around “weekly”)	0.01 (0.03)	[−0.04; 0.08]	−0.04 (0.04)	[−0.13; 0.03]
Milk (centered around “monthly”)	−0.03 (0.04)	[−0.11; 0.05]	0.05 (0.03)	[−0.00; 0.11]
Tea (centered around “monthly”)	0.05 (0.04)	[−0.03; 0.13]	0.04 (0.06)	[−0.07; 0.16]
Cocoa (centered around “monthly”)	0.02 (0.03)	[−0.04; 0.09]	0.07 * (0.03)	[0.00; 0.15]
**School-level predictors**				
Locality type (−1 Rural, 1 Urban)	−0.11 (0.06)	[−0.29; −0.04]	−0.00 (0.05)	[−0.12; 0.10]
County development index (centered around the mean)	−0.00 (0.00)	[−0.01; 0.01]	−0.00 * (0.00)	[−0.01; −0.00]

* *p* < 0.05; ** *p* < 0.01.

## Data Availability

The data presented in this study are available on request from the corresponding author. The data are not publicly available in accordance with consent provided by participants on the use of confidential data.

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
