# Peer review of "Caries Prevalence Associated with Oral Health-Related Behaviors among Romanian Schoolchildren"

_ijerph, 2021, doi:10.3390/ijerph18126515_

Round 1
Reviewer 1 Report
Authors have responded to my earlier comments, however, not registering the participation rate can be considered as one of the limitations of this study.
Author Response
Dear reviewer,
Thank you very much for your comments. The authors added the fact that not registering the participation rate can be considered a limitation of the study.

Reviewer 2 Report
The reviewer would like to thank the authors for their efforts. The revised manuscript after resubmission addressed all concerns of the reviewer.
Author Response
Dear reviewer,
Thank you very much for your comments and for your time.
This manuscript is a resubmission of an earlier submission. The following is a list of the peer review reports and author responses from that submission.
Round 1
Reviewer 1 Report
The manuscript entitled “Caries prevalence associated to oral health related behaviors among 12 years old Romanian schoolchildren” aimed to evaluate caries prevalence and severity among 12-year-olds children, residing in rural and urban areas of Romania, and to correlate it with oral health related behaviors. I have few comments that the authors may take into consideration.
- Authors have missed some of the important papers in the literature review. There are some previously published scientific papers concerning the dental caries status among children/adolescents in Romania.
- Can authors elaborate the need of ‘National Oral Health Survey’?
- On what basis, did the authors categorize the locality type as urban vs rural? How many schoolchildren were approached and what was the response rate? Did the authors managed to fulfil the sample size determined?
- This survey was conducted between 2019-2020, were the examiners recalibrated during this period? Did authors perform re-examination?
- Can authors describe the oral health-related behaviours variable in the method section?
- In the title and methods, it says that this study was conducted among 12-year-olds, however, authors have presented schoolchildren aged 11–14-year-olds in the result section. Can authors justify this?
- Can authors compute and present the confidence intervals for regression analyses? Were models adjusted for confounders?
- Can authors explain the county developmental index and RT index?
- Please follow the STROBE checklist for discussion section. Discussion lacks the implication of study findings.
Author Response
Dear Reviewer,
Thank you very much for your comments. You can find the response to your questionsin the attached file

Reviewer 2 Report
The manuscript by Ruxandra Sava-Rosianu et al. presents an interesting study valuate caries prevalence, severity and associated factors in 12 years old children, residing in rural and urban areas of Romania. The study is well designed and written and the conclusions are well-supported by the results. Nevertheless, some improvements can be performed before the manuscript is published.
1- Line 26: Please change "condition" to "conditions" or "play" to "plays"
2- Line 44: Please make sure to use MESH keywords
3- Line 77: Why specifically 12 years old children? Could there be a difference to other age groups? Please explain and discuss.
4- Line 115: What are the special lightening and examination kits? Please explain more.
5- Line 247-248: What is the effect of gender on the same topic in other countries? Please compare.
6- Line 267: Who is more effective, mothers or fathers and why? Please explain.
Author Response
Dear Reviewer,
Thank you very much for your comments. You can find the response to your comments in the attached file.

Round 2
Reviewer 1 Report
Authors have responded to my earlier comments, however, most of them are not presented in the manuscript. Therefore, I suggest authors to amend the current form of manuscript.
I suggest authors to highlight/discuss on the following topics:
- categorization of locality type, county’s developmental index and RT index. International readers may not be aware of these variables; therefore, it’s worthwhile to explain these in methods section.
- participation rate (how many invited for the study and how many participated. If not calculated, please mention this), re-examination not performed (see page 26-27, WHO Oral Health Surveys – Basic Methods, 5th edition, 2013).
- Authors have examined schoolchildren of 11–14-year-olds, but the title suggests that the participants were only 12-year-olds. Therefore, I suggest authors making a minor change in both title and methodology section.
- Do you think it is necessary to adjust the potential confounders in the regression model? Your covariates are socioeconomic, and oral health-related behaviours.
- Authors have now brought an important topic ‘complex sampling’, how did you take this into consideration during analyses? (suggest reading Caplan, D.J., Slade, G.D. and Gansky, S.A. (1999), Complex Sampling: Implications for Data Analysis. Journal of Public Health Dentistry, 59: 52-59. https://doi.org/10.1111/j.1752-7325.1999.tb03235.x)
Author Response
Dear Reviewer,
Thank you for your comments. You can find the responses to your questions in the attached document.
